# Telomere Function and the G-Quadruplex Formation are Regulated by hnRNP U

**DOI:** 10.3390/cells8050390

**Published:** 2019-04-28

**Authors:** Hiroto Izumi, Keiko Funa

**Affiliations:** 1Department of Occupational Pneumology, Institute of Industrial Ecological Sciences, University of Occupational and Environmental Health, Fukuoka 807-8555, Japan; h-izumi@med.uoeh-u.ac.jp; 2Department of Medical Biochemistry and Cell Biology, Institute of Biomedicine, University of Gothenburg, SE-40530 Gothenburg, Sweden; 3Oncology Laboratory, University of Gothenburg, Sahlgrenska Hospital, Gula stråket 8, SE-41345 Gothenburg, Sweden

**Keywords:** telomeres, hnRNP U, G-quadruplex, ssTel-oligonucleotide, Tel-oligonucleotide, RPA

## Abstract

We examine the role of the heterogenous ribonucleoprotein U (hnRNP U) as a G-quadruplex binding protein in human cell lines. Hypothesizing that hnRNP U is associated with telomeres, we investigate what other telomere-related functions it may have. Telomeric G-quadruplexes have been fully characterized in vitro, but until now no clear evidence of their function or in vivo interactions with proteins has been revealed in mammalian cells. Techniques used were immunoprecipitation, DNA pull-down, binding assay, and Western blots. We identified hnRNP U as a G-quadruplex binding protein. Immunoprecipitations disclosed that endogenous hnRNP U associates with telomeres, and DNA pull-downs showed that the hnRNP U C-terminus specifically binds telomeric G-quadruplexes. We have compared the effect of telomere repeat containing RNA (TERRA) on binding between hnRNP U and telomeric (Tel) or single- stranded Tel (ssTel) oligonucleotides and found that ssTel binds stronger to TERRA than to Tel. We also show that hnRNP U prevents replication protein A (RPA) accumulation at telomeres, and the recognition of telomeric ends by hnRNP suggests that a G-quadruplex promoting protein regulates its accessibility. Thus, hnRNP U-mediated formation has important functions for telomere biology.

## 1. Introduction

Telomeres are nucleoprotein complexes that constitute the natural ends of eukaryotic chromosomes. One strand is G-rich and forms a single-stranded 3′ overhang, whereas the complementary strand is C-rich and has a recessed 5′ end [1]. Telomeric DNA is highly conserved between species (vertebrates) with an identical TTAGGG repeat sequence [2]. The essential role of telomeres is to protect chromosome ends from recombination and fusion and from being recognized as broken DNA ends. This is achieved by telomere-binding proteins. Heterogenous ribonucleoprotein U (hnRNP U) has been reported to be part of a telomerase complex, but its functions in telomeres has not been precisely known, necessitating further studies. hnRNP U has been suggested to be a telomere-end binding protein, contributing to telomere length regulation [2]. Replication of linear chromosomes fails to make a complete copy of the lagging strand due to discontinuous synthesis and the need for RNA priming, known as the end replication problem [3]. Thus, the length of the telomeric DNA gradually shortens as cells divide.

In order to balance DNA loss at chromosome termini, telomerase, being a ribonucleoprotein (RNP) reverse transcriptase, adds back telomeric single-stranded DNA (ssDNA) repeats by iterated reverse transcription of a template sequence within its intrinsic RNA, as long as the telomeric DNA remains an appropriate substrate. Telomere extension depends on the conformation of the telomeric single-stranded 3′ overhangs and their folding into secondary structures, known as G-quadruplexes [4,5,6,7,8], will make telomerase unable to extend telomeres. The G-quadruplex is formed in nucleic acids, which are rich in guanines, and it is known to have a secondary structure. Critical factors include how telomeric 3′ overhangs are folded into G-quadruplexes, stopping the extension of telomeres by telomerase. There are other structures that protect telomeres, including T-loops and the G-rich overhang, which covers the double-strand part of the telomere, forming a so-called T-loop. There exists also a smaller loop where the complementary G-rich strand is displaced, a so-called D-loop.

Moreover, telomeric G-quadruplexes also prevent initiation of DNA damage responses in cell extracts [9]. In addition, two single-stranded DNA (ssDNA)-binding proteins—replication protein A (RPA) and protection of telomere 1 (POT1)—play critical roles in DNA replication and telomere capping, respectively [10]. RPA is involved in DNA repair and other DNA processes and binds telomeric ssDNA. However, RPA must be removed following completion of replication for POT1 in order to bind telomeric ssDNA [11]. POT1 is one of the components of the shelterin complex that binds to the TTAGGG repeats of telomeres, controlling telomere length and protecting chromosome ends from abnormal recombination, chromosome instability, and segregation. The POT1 and POT1- interacting protein 1 (TPP1) binds telomeric ssDNA and protects G overhangs against RPA binding [12]. hnRNP A1 has been known to bind G-quadruplexes and telomeres [13]. hnRNP A1 can remove RPA from telomeric ssDNA and is itself displaced by a telomere repeat containing RNA (TERRA), promoting an RPA-to-POT1/TPP1 to switch [14].

We find that hnRNP U prevents RPA from recognizing telomeric ends, and also that hnRNP U-mediated formation of G-quadruplexes carries important functions. hnRNP U has been reported to be part of a telomerase complex negatively regulating telomere length, transcriptional regulation, and controlling X inactivation [15,16]. The hnRNP U contains an arginine–glycine–glycine (RGG) domain, which in other proteins have been reported to facilitate the formation of G-quadruplexes [17]. TERRA has been shown to stabilize G-quadruplexes more than those formed by telomeric DNA [18] and will thus compete with telomeric DNA for hnRNP U binding. The hnRNP U is a major scaffold/matrix attachment region (S/MAR) binding protein, and telomeres have been known to constitute S/MARs [16]. There are several more unresolved functions in telomere biology, and thus we wanted to study the possible roles of hnRNP U.

## 2. Materials and Methods

### 2.1. Cell Culture

COS1 cells (monkey kidney fibroblast-like cell line) were grown in Dulbecco’s modified Eagle medium (DMEM) containing 8% fetal bovine serum (FBS). mES R1 cells were grown either under feeder-free conditions in 15% EmbryoMax FBS, containing DMEM media supplemented with ESGRO recombinant mouse leukemia inhibitory factor (LIF) protein, according to manufacturer’s instructions (Millipore, PB4090EN00), or on a layer of Mitomycin C inactivated mouse embryonic fibroblasts (MEFs) [19]. All other cell lines were grown in DMEM supplemented with 10% FBS, ultra-glutamine, and penicillin–streptomycin. These cells were grown at 37 °C under 5% CO_2_.

### 2.2. Plasmid Constructs and Transfections

The full length of hnRNP U complementary DNA (cDNA) was obtained from Human Universal QUICK-Clone II (Clontec) as a template. With this cDNA, Flag-N704, Flag-200-472, Flag- 474-704, and Flag-683C were obtained by specific restriction enzymes, as described by us.

For construction of Flag-hnRNP U N704 we used MlsI restriction enzyme site. For bacteria expression, the pThioHis vector-deleted ThioHis (TH) sequence was used [20]. cDNA fragments of hnRNP U 200/472 and 474/704 were constructed by using EheI, Bst1107I, and MlsI restriction enzyme sites, respectively. A cDNA fragment of hnRNP U 683C was obtained with primer pairs

5′—CAGAACACTGGCTCAAAGAAAAGCAATAAA—3′ and 5′—TCAATAATATCCTTGTGATAAT CTGACT—3′).

These cDNA fragments were Flag-tagged at the N-terminal and were ligated into the TH vector. For transfection of expression plasmids, COS1 cells were seeded into a six-well plate with 5 µg of plasmid and 10 µL of X-tremeGENE9 DNA. Transfection reagents were used according to instructions, and transfectants were selected by 3 µg/mL puromycin.

### 2.3. Recombinant Protein Expression and Purification

Flag-hnRNP U 200/472, 474/704, and 683C were expressed in *E. coli* DH5α for 1 h with 1 mM isopropyl-β-tiogalactoside (IPTG). Cells were collected by centrifugation and sonicated for 30 s in lysis buffer containing 50 mM Tris–HCl (pH 8.0), 1 mM EDTA, 120 mM NaCl, 0.5% Nonidet P-40, and 0.5 mM phenylmethylsulfonyl fluoride (PMSF), and centrifuged at 21,000 ×*g* for 10 min at 4 °C. The supernatants (10 mg bacteria) were incubated with 10 μL anti-Flag M2-agarose affinity gel for 30 min at 4 °C. The gels containing Flag-hnRNP U fusion protein were washed with buffer containing 100 mM KCl, 10 mM Tris-HCl pH 7.4, 0.05% NP-40, and 10% glycerol. The Flag-hnRNP U fusion protein was used in each assay. In dissociating *E. coli* DNA, the beads were incubated with 0.4 M NaCl, 10 mM Tris-HCl pH 7.4, 0.05% NP-40, and 10% glycerol for 30 min at 4 °C, and then washed. The COS1 transfectant expressing Flag-hnRNP U FL and N704 was collected by centrifugation. Each cell was separated into nucleus and cytoplasm as described [21]. The nuclear fraction was used for immunoprecipitation of Flag- hnRNP FL and N704, including the nuclear localization signal [22]. Each fraction (100 µg) was incubated with 10 μL anti-Flag M2-agarose gel for 30 min at 4 °C, and the gels containing Flag-hnRNP U fusion protein were washed.

### 2.4. Competition Assay with E. coli DNA

Flag-hnRNP U proteins were expressed in COS1 cells and extracted from the nucleus, as described above. Flag-hnRNP U was incubated with indicated biotin-linked oligonucleotides with KCl buffer for 30 min at room temperature (RT) and washed three times with KCl buffer. Bound oligonucleotides were dissociated with 2 M NaCl for 30 min at RT. After centrifugation at 21,000 rpm for 10 min, oligonucleotides in supernatant were transferred to a polyvinylidene difluoride (PVDF) membrane by HYBRI-SLOTTM Manifold. Blotted biotin-linked oligonucleotides were detected by a streptavidin-horseradish peroxidase (HRP) conjugate. Images were obtained using an analyzer (LAS-4000 mini, Fujifilm, Tokyo, Japan). In order to compare the effects of KCl and LiCl, the binding activity between Flag-hnRNP U full-length and telomeric (Tel) oligonucleotide was performed, replacing 100 mM KCl of binding buffer and then washing the buffer with 100 mM LiCl. To analyze the effects of DNA on binding hnRNP U and Tel oligonucleotide, indicated amounts of purified *E. coli* DNA were added to the binding buffer containing Flag- hnRNP U fusion protein.

### 2.5. Effect of TERRA on Binding between hnRNP U 683C and Tel or Single-stranded(ss)Tel Oligonucleotide

*E. coli*-derived Flag-hnRNP U 683C was immunoprecipitated with anti-Flag M2-agarose affinity gel. After washing, agarose gels, 1 µL of 10 µM Tel 4.01 or ssTel oligonucleotides with indicated volume of 10 µM TERRA oligonucleotide, were mixed in KCl buffer. After incubation for 30 min at RT, 1 µL of 20 mg/mL proteinase K was added and incubated for 15 min at 37 °C, followed by 10 min at 80 °C. After centrifugation with 12,000 rpm for 1 min, the supernatant was purified with a 0.45 µm filter by centrifugation. The flow-through was mixed with 10 µL streptavidin-DynaBeads in KCl buffer and mixed for 15 min at RT and then washed once more with KCl buffer. Tel linker binding to the 3′ single-strand lesion of Tel or ssTel oligonucleotides, and KCl buffer were added and mixed for 10 min at RT. The bound complex was washed 3 times with KCl buffer, and 200 µL 0.1× TE buffer was added. After vortex, qPCR based on SYBR-Green fluorescence was performed with 1 µL of sample and primer pairs (Linker primer 1 and 2). DNA quantification was calculated by generating a standard curve from a dilution series. Each DNA amount without TERRA competition was set to 1. Each sample was duplicated.

### 2.6. Western Blotting

Proteins derived from COS1 cells or *E. coli* were subjected by SDS-PAGE and transferred to PVDF membrane. Flag and RPA2 were detected with specific 1st antibodies and bound 2^nd^ antibodies were visualized using an enhanced chemiluminescence kit (GE Healthcare Bio-Sciences, Pittsburgh, PA, USA). Biotinylated oligonucleotides were transferred to PVDF membrane by HYBRI-SLOTTM Manifold. Bound streptavidin-HRP was visualized as described above.

### 2.7. Exonuclease I Protection Assay

*E. coli*-derived Flag-hnRNP U 683C was immunoprecipitated with anti-Flag M2-agarose affinity gel, as described above. After washing the agarose gel, 1 µL of 10 µM T24G21 oligonucleotide or T24RG21 oligonucleotide was mixed in KCl buffer. After incubation for 30 min at RT, agarose affinity gels were washed twice with KCl buffer and incubated with Exonuclease I in 100 µL appropriate buffer for 15 min at 37 °C. Next, 10 µL of streptavidin-DynaBeads and 800 µL of KCl buffer were added and mixed for 15 min at RT. The complexes were washed twice with KCl buffer and incubated with 1 µL of 10 µM T24G21 linker or T24RG21 linker for 15 min at RT. Then complexes were washed three times, and finally 200 µL 0.1× TE buffer was added in the tube. DNA quantification of each linker was performed by qPCR as described above with same primer pairs (Linker primer 1 and 2). Each DNA amount without Exonuclease I was set to 1. Each sample was duplicated.

### 2.8. huRNP U Prevents RPA2 from Binding Telomeres

*E. coli*-derived Flag-hnRNP U 683C was immunoprecipitated with anti-Flag M2-agarose affinity gel, as described above. After the removal of *E. coli*-derived DNA, complexes were mixed with 20 µg of mouse embryonic stem cell extract and whole cell extract (WCE) and 10 µL of 10 µM ssTel or ssTel(da) for 3 h at 4 °C. After centrifugation at 2000 rpm for 5 min at 4 °C, supernatants were transferred into a new tube and mixed with 20 µL streptavidin-DynaBeads and mixed for 1 h at 4 °C and washed three times with KCl buffer. Resulting pellets were subjected to Western blotting with anti-RPA2 antibody, as described above.

### 2.9. Oligonucleotides

Acrylamide gels with purified oligonucleotides as listed below were ordered from GeneLink. Folding of hairpins was evaluated using the Mfold online software [19].

Tel 4.01: 5′—CTAACCCTAA CCCTAACCCT AACCCTAACC CTAACCCTAA CCCTAACCCT AAACT***X***CGGT TTAGGGTTAG GGTTAGGGTT AGGGTTAGGG TTAGGGTTAG GGTTAGGGTT AGGGTTAGGG TTAGGGTTAG GGTTAGGGTT AGGGTTA*G*—3′Tel(da): 5′—CTAACCCTAA CCCTAACCCT AACCCTAACC CTAACCCTAA CCCTAACCCT AAACT***X***CGYT TTAGYGTTAY GYTTAGYGTT AYGYTTAGYG TTAYGYTTAG YGTTAYGYTT AGYGTTAYGY TTAGYGTTAY GYTTAGYGTT AYGYTTA*G*—3′Tel(ds): 5′—CTAACCCTAA CCCTAACCCT AACCCTAACC CTAACCCTAA CCCTAACCCT AAACT***X***CGGT TTAGGGTTAG GGTTAGGGTTA GGGTTAGGGT TAGGGTTAGG GTTAGGGTTA G—3′Cnt: 5′—TGGCATATCA CGGTCGTGCG CGTAGAGATG TGATCGGAGA AAAGAGTCAT TACT***X***CGGTA ATGACTCTTT TCTCCGATCA CATCTCTACG CGCACGACCG TGATATGCC AGCAAGACGC TCTGCCTCGC TGTTGATGAT GTTTCGAA—3′ssTel: 5′—***Z***GGTTAGGGT TAGGGTTAGG GTTAGGGTTA GGGTTA*G*—3′ssTel(da): 5′—***Z***YGTTAYGYT TAGYGTTAYG YTTAGYGTTA YGYTTA*G*—3′ssCnt: 5′—***Z***CAAGACGCTC TGCCTCGCTG TTGATGATGT TTCGAA—3′TERRA: 5′—GGUUAGGGUU AGGGUUAGGG UUAGGGUUAG GGUUAG—3′dGGG(TTAGGG)_3_: 5′—***Z***GGGTTAGG GTTAGGGTTA GGG—3′TelC: 5′—CTAACCCTAA CCCTAACCCT AACC***Z***—3′

***X*** = Biotin dT; ***Z*** = Biotin TEG; G = enzymatically (T4 TdT, New England Biolabs) added ddG (GE Life Science) for all experiments; Y = 7-deaza-8-aza-dG.

The following gel purified oligonucleotides were ordered from MWG Eurofines:T24G21: 5′—Biotin-T_24_(G_3_T_2_A)_3_G_3_—3′T24RG21: 5′—Biotin-T_24_GTGTGAGTGGAGGTGTGAGGT—3′Tel linker: 5′—GGGCTGGCAA GCCACGTTTG GTGTAAAACG ACGGCCAGTA GAAGGCACAG TCGAGGCCTC TGACACATGC AGCTCCCGGC TAACCCTAAC CCTAACCCT—3′T24G21 linker: 5′—GGGCTGGCAA GCCACGTTTG GTGTAAAACG ACGGCCAGTA GAAGGCACAG TCGAGGCCTC TGACACATGC AGCTCCCGGC CCTAACCCTA ACCCTAACCC—3′T24RG21 linker: 5′—GGGCTGGCAA GCCACGTTTG GTGTAAAACG ACGGCCAGTA GAAGGCACAG TCGAGGCCTC TGACACATGC AGCTCCCGGA CCTCACACCT CCACTCACAC—3′Linker primer 1: 5′—GGGCTGGCAA GCCACGTTTG GTG—3′Linker primer 2: 5′—CCGGGAGCTG CATGTGTCAG AGG—3′

### 2.10. Antibodies

The antibodies were used at the indicated concentrations for Western blotting: Mouse monoclonal to Flag (Sigma M2); 1:5000. Rabbit polyclonal to RPA (abcam, ab97594); 1:1000.

## 3. Results

### 3.1. hnRNP U Associates with Telomeres

In order to investigate whether hnRNP U associates with telomeres, we made Flag-hnRNP U full-length (FL) and Flag-hnRNP U N704 (expressing 1-704 amino acids) fusion proteins expressed by COS1 cells (Figure 1A). These proteins were mixed with biotinylated Tel 4.01 oligonucleotide. Bound oligonucleotides were transferred to membrane and detected with streptavidin-HRP using electrochemical luminescence (ECL). The binding assay showed that hnRNP U FL bound Tel 4.01, but hnRNP U N704 did not, suggesting that the C-terminus of hnRNP U binds Tel 4.01 and G-quadruplex (Figure 1B). The same binding assay was done with Flag-hnRNP U FL with Tel 4.01 under G-quadruplex promoting (100 mM KCl) and inhibiting (100 mM LiCl) conditions.

Three different constructs of Flag-hnRNP are illustrated in Figure 2A. Figure 2B, shows Flag- hnRNP U 683C (683-825 amino acid [aa]) to bind Tel 4.01, but Flag-hnRNP U 200/472 (200-472 aa) and Flag- hnRNP U 474/704 (474-704 aa) did not. These results are consistent with Figure 1A. As expected, LiCl buffer inhibited the binding between Flag-hnRNP U FL and Tel 4.01, without disturbing the binding between anti-Flag and Flag-hnRNP U (Figure 1C).

We made three different constructs of Flag-hnRNP U protein expressed in *E. coli* located in the C-terminal region. In the following experiments, Flag-hnRNP U FL and Flag-hnRNP U 683C fusion proteins were used, expressed by COS1 cells and *E. coli*, respectively.

### 3.2. Binding Assays of Flag-hnRNP U and Several Telomere-Associated Oligonucleotides in Competition with E. coli DNA

We synthesized biotinylated oligonucleotides for the binding assay, containing both double- stranded (ds) and single-stranded (ss) telomeric DNA (Tel), and control DNA (Cnt) (Figure 3A). Assays could contain either ss or ds on its own, plus a non-G-quadruplex forming a 7-deaza-8-aza dG-substituted(da) oligonucleotide. These oligonucleotides were mixed for competition with Flag- hnRNP U FL fusion proteins, expressed by COS1 and indicated amounts of *E. coli* DNA. Bound oligonucleotides were transferred to membrane and detected with streptavidin-HRP using ECL (Figure 3B). Flag-hnRNP U FL fusion proteins could bind to these oligonucleotides, including control oligonucleotides. Competition assay revealed that hnRNP U preferentially binds single stranded 3′ overhangs when compared to ssDNA or dsDNA on their own, and further shows that in double-stranded form, the Tel(da) oligonucleotide was unable to form Hoogsten basepairs and had much better binding than the Tel(ds) oligonucleotide, which can form G-quadruplex.

### 3.3. Binding Assays of Flag-hnRNP U683C and Tel or ssTel. Competition with TERRA

Flag-hnRNP U 683C fusion protein made by *E. coli* was mixed with Tel 4.01 or ssTel, including indicated excess mole TERRA. Bound Tel 4.01 were pulled down by streptavidin magnetic beads and mixed with Tel linker oligonucleotide. Tel linker bound to Tel 4.01 was used as a template for qPCR. As shown in Figure 4, TERRA inhibited the binding between Flag-hnRNP U 683C and Tel 4.01 more efficiently than that between Flag-hnRNP U and ssTel. These results suggested that TERRA competes for binding, but the preferred substrate is telomeric DNA. Moreover, it suggests that the hnRNP U 683 C has a higher affinity for telomeric ssDNA than for TERRA.

### 3.4. Flag-hnRNP U 683 C Facilitates G-Quadruplex Formation and DNA Degradation

We tested an exonuclease protection assay in order to evaluate any G-quadruplex-promoting stability by hnRNP U 682 C. To investigate the protection of DNA degradation by Flag-hnRNP U, Flag-hnRNP U 683C fusion proteins were mixed with biotinylated oligonucleotides, a G-quadruplex-forming (T24G21) oligonucleotide, or a control oligonucleotide unable to form G-quadruplex (T24RG21). Bound oligonucleotides were treated with Exonuclease I and pulled down by streptavidin magnetic beads. T24G21 linker or T24RG21 linker bound to T24G21 or T24RG21, respectively, which were used as a template for qPCR. As shown in Figure 5, the T24RG21 oligonucleotide was mostly degraded, but the degradation of the T24G21 oligonucleotide was only partial. This result indicated that the hnRNP U 683 C, but not the control (Exol-), would protect the G-quadruplex formation from Exonuclease I.

### 3.5. huRNP U Prevents RPA2 Accumulation on Telomeres

It has been suggested that G-quadruplex formation can prevent RPA2 from binding ssTel and induce a DNA damage response [23]. Does hnRNP U prevent RPA2 from binding to telomeric ends? It has also been proposed that telomeres form during replication. We have made a Flag- hnRNP U 683C fusion protein from which the *E. coli* genomic DNA was removed. It was mixed with mES WCE and ssTel or ssTel(da) oligonucleotides. RPA2-bound Flag-hnRNP U 683C was subjected to Western blot with anti-RPA2 antibody. As shown in Figure 6, ssTel binds to hnRNP U 683C but not to ssTel(da). This result suggests that hnRNP U683C competes with the binding of RPA2 and promotes binding of ssTel.

## 4. Discussion

We report that hnRNP U will associate with telomeres in COS1 cell line. We aimed to understand whether hnRNP U promotes G-quadruplex activity. The area of hnRNP U that binds to telomeres was shown to be its glycine-rich C-terminal RGG-domain—binding telomeric DNA and increasing the formation of G-quadruplexes. This binding was not observed when using the same assay in a LiCl condition that inhibits G-quadruplexes.

G-quadruplexes have been thought to be formed at telomeric ends. Several reports state that helicases are involved in their unfolding [23,24]. It could be that hnRNP U supports G-quadruplex formation in order to control the necessary proteins involved in telomere replication and protection, which would affect RPA binding. Our studies only partially support the in vitro data, showing that hnRNP U will directly prevent RPA accumulation at telomeres, but the in vivo situation may be more intricate. Our finding that hnRNP U prevents RPA from binding to telomeric ends would suggest that a G-quadruplex promoting protein controls the accessibility of telomeric ssDNA.

We found that hnRNP U associates with telomeres, promoting the formation of telomeric G-quadruplexes. The importance of G-quadruplexes in genome stability has been established through characterization of helicases that unfold them [19,25]. hnRNP U has previously been reported to be part of a telomerase complex, negatively regulating telomere length [26]. In addition, hnRNP U is directly associated with telomeres. It could be that hnRNP U controls the accessibility of telomeric 3′ overhangs to telomerase. Our results show that hnRNP U-mediated formation of G-quadruplexes plays important roles in telomere biology.

hnRNP U is one major structural constituent of the nuclear matrix and might have broader roles in telomere biology. Telomeres are known to be S/MARs, located at the distal ends of chromosomes [27], and they are equipped with single stranded 3′ overhangs. S/MARs tether the chromatin to the nuclear matrix, where transcription is going on and poising the chromatin for transcription [15,28]. The nuclear matrix specifically recognizes telomeric ssDNA, and the hnRNP U has been reported to be part of a telomere complex [29]. It is thus possible that by folding and unfolding of G-quadruplex, hnRNPs will control the accessibility of telomeric ssDNA at the nuclear matrix, setting the attachment of telomeres to specific matrix areas. Thus, hnRNP U may tether fixed points of a telomere to the nuclear matrix to establish transcriptionally active chromatin loops. In fact, the hnRNP U is also known to facilitate RNA pol II-mediated transcription and the formation of chromatin loops [15,28]. Recently, G-quadruplexes have been discovered throughout mammalian genomes at telomeres and other locations [7,8].

## 5. Conclusions

The protein hnRNP U promotes the formation and activity of G-quadruplexes. hnRNP U has several important functions—it binds to telomeric G-quadruplex and prevents RPA from recognizing telomeric ends. Also, hnRNP U is a major structural constituent of the nuclear matrix and it is likely to play a broad role in telomere biology.

## Figures and Tables

**Figure 1 cells-08-00390-f001:**
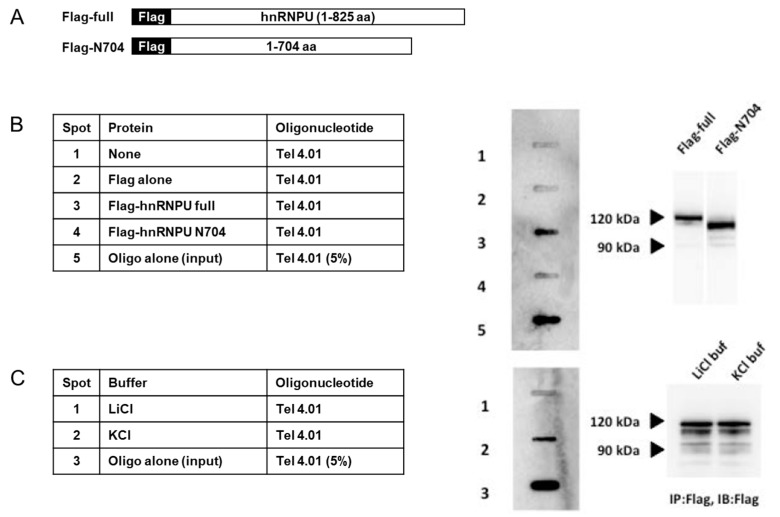
(**A**) Schematic illustration showing full length and N704 (expressed 1-704 aa) of Flag- heterogenous ribonucleoprotein U (hnRNP U) protein. (**B**) Flag-hnRNP U full length and N704 fusion proteins expressed by COS1 cells were immunoprecipitated (IP:d) with anti-Flag agarose gel and were mixed with Tel oligonucleotides in KCl buffer. After washing the beads, bound oligonucleotides were transferred to the membrane and detected by electrochemical luminescence (ECL). (**C**) Flag-hnRNP U FL-fusion proteins extracted from COS1 cells were IP:d with anti-Flag agarose gel. IP were mixed with Tel in KCl buffer or LiCl buffer. Same treatment as in Figure 1A. Oligo alone (input) indicates that the 5% oligo used in the binding assay was spotted directly.

**Figure 2 cells-08-00390-f002:**
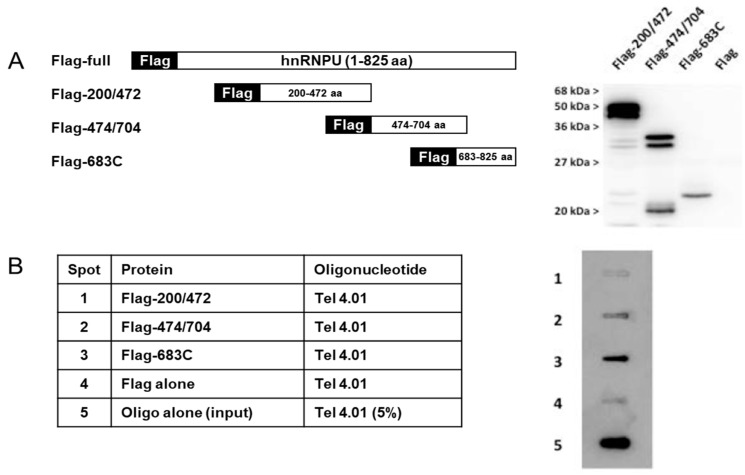
(**A**) Schematic illustration showing three different constructs of Flag-hnRNP U protein made in *E. coli*, and each protein was subjected by Western blotting with anti-Flag antibody. (**B**) Each Flag-hnRNP fusion protein was IP:d with anti-Flag agarose gel, and IP were mixed with Tel in KCl buffer. Same treatment as in Figure 1A. Oligo alone (input) indicates that the 5% oligo used in the binding assay was spotted directly.

**Figure 3 cells-08-00390-f003:**
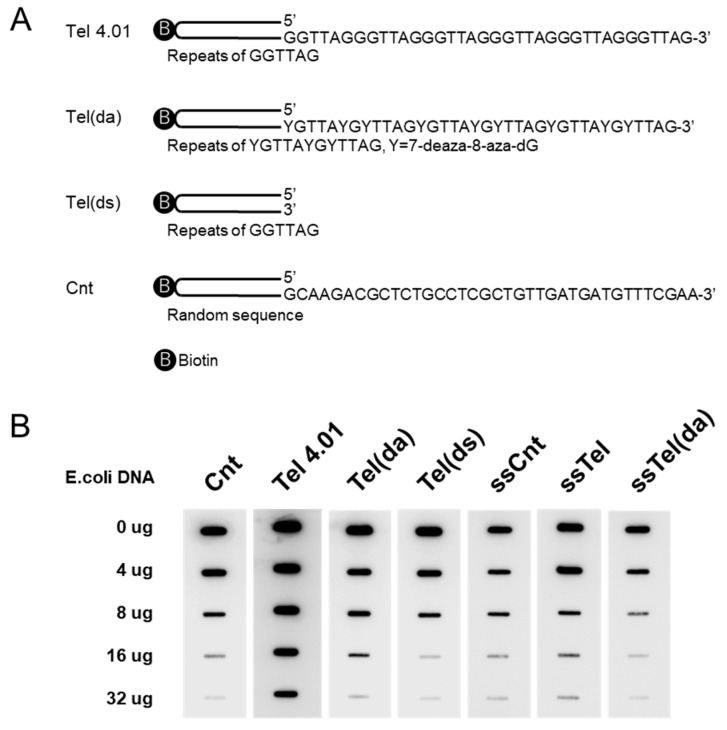
(**A**) Schematic illustration showing double-strand oligonucleotides. Full base sequences of each oligonucleotide are shown in Materials and Methods section. (**B**) Flag-hnRNP U FL fusion proteins extracted from COS1 cells were IP:d with anti-Flag agarose gel. IP were mixed with indicated oligonucleotides, adding various weights of *E. coli* DNA in KCl buffer. Same treatment as in Figure 1A.

**Figure 4 cells-08-00390-f004:**
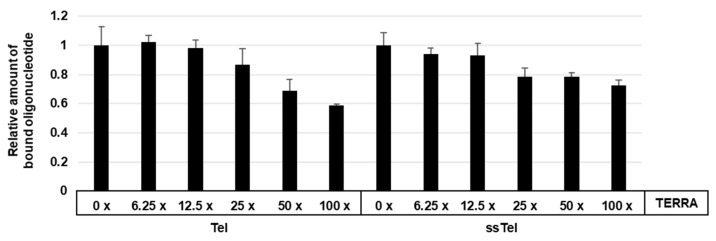
Flag-hnRNP U 683C fusion proteins made in *E. coli* were IP:d with anti-Flag agarose gel. IPs were mixed with Tel or ssTel oligonucleotides, followed by adding indicated-fold amount of TERRA oligonucleotides. After washing the beads, bound oligonucleotides were purified and pulled down by streptavidin magnetic beads. After washing the beads, adapter oligonucleotide (Tel linker) was mixed and bound adapter oligonucleotide was detected by qPCR. See Appendix A.

**Figure 5 cells-08-00390-f005:**
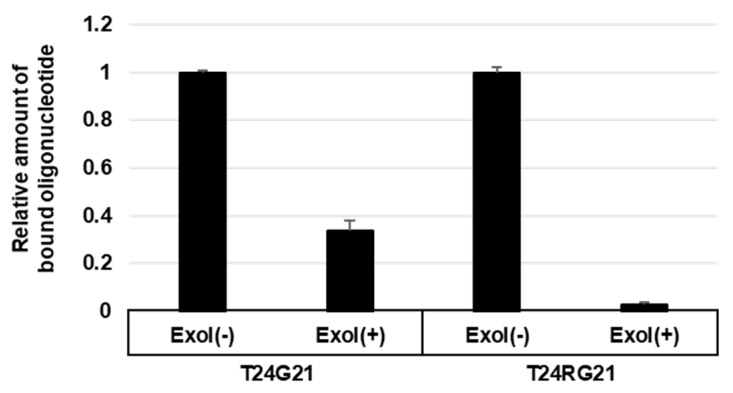
Flag-hnRNP U 683C fusion proteins made in *E. coli* were IP:d with anti-Flag agarose gel. IP were mixed with T24G21 and T24RG21. After washing the beads, exonuclease was added and incubated. After washing the beads, bound oligonucleotides were purified and were pulled down by streptavidin magnetic beads. After washing the beads, adapter oligonucleotide (T24G21 linker or T24RG21 linker) was mixed and bound adapter oligonucleotides were detected by qPCR. See Appendix A.

**Figure 6 cells-08-00390-f006:**
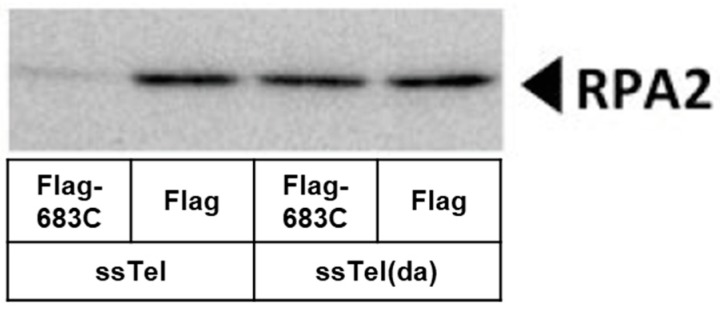
hnRNP U prevents RPA2 from binding telomeres. Flag-hnRNP U 683C fusion proteins made in *E. coli* were IP:d with anti-Flag agarose gel. Genome DNA derived from *E. coli* was removed from IP and mixed with mouse embryonic stem cell extract and whole cell extract (WCE) and ssTel or ssTel(da) oligonucleotides. After centrifugation, each oligonucleotide in the supernatants was pulled down with streptavidin magnetic beads. RPA2-bound oligonucleotides were detected by Western blot with anti-RPA2 antibody.

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
