# Peer review of "Telomere Function and the G-Quadruplex Formation are Regulated by hnRNP U"

_cells, 2019, doi:10.3390/cells8050390_

Round 1

Reviewer 1 Report

Izumi and Funa reported the telomere-related function of the hnRNP U protein with respect to its binding to telomere oligonucleotides and stabilization of G-quadruplex structures. They showed that the C-terminal part of the protein is crucial for the specific binding of telomere quadruplex structures. Next, they presented that hnRNP U binding to telomeres prevented RPA2 accumulation on these structures. Although authors dealt with the promising and generally interesting research topic, the data suffer from many drawbacks complicating their understanding and making it difficult to find clear and unambiguous acquisition of this work This statement can be stressed, e.g., by the final sentence of the Abstract: “It is necessary to uncover possible roles of hnRNP U in telomere biology” – in my opinion, to contribute to this was the main goal of the work.

Specific comments:

·         There was stated in the Abstract (l. 17-18) that “new methods confirmed by the standard method” were used. Which of methods listed (l.18-19) are considered as new and which as standard ones?

·         Conclusions in Abstract, l. 21-24, should be re-phrased to be clear per se (without reading results).

·         When the sentence in the Intro is introduced by “We have suggested that…” it is expected that the work of authors of the ms will be presented. This is not the case here, l. 38-40.

·         G-quadruplexes are presented as structural motives regulating access of telomeres for telomerase (l. 44-46). In this context, other secondary telomere DNA structures, as t-loops and D-loops, should be mentioned.

·         How cDNA fragments were constructed by restriction enzymes digestion (l. 77-79), what was the template for this digestion?

·         RT-PCR (l. 122) is a standard abbreviation for reverse transcription followed by PCR. For quantitative PCR, I would recommend qPCR.

·         I was rather confused by the structure of oligonucleotides used for binding assays (Chapter 2.9.). Am I right that Tel4.01 is ds oligo with 3´overhang, Tel(ds) ds oligo without 3´overhang and Tel(da) ds oligo containing 7-deaza-8-aza-dG with 3´overhang? I would recommend to add this information when introducing respective experiment in the Results section.

·         There is stated in the text that LiCl-containing buffer (G4-inhibiting conditions) did not inhibit the binding between anti-Flag and Flag-hnRNP U (199-200) but there is no comment on the loss of binding of Tel4.01 and Flag-hnRNP U FL under these conditions, which is, in my opinion, the crucial result of this experiment.

·         The statement “Oligo alone” in Figure 1 tables is rather confusing because interpretation that oligo was used in the assay instead of respective protein is possible.

·         Authors compared binding of Tel(da) and Tel(ds) oligos to Flag-hnRNP U FL (l. 230-231, Figure 3). If I understood well the structural specificities of respective oligos, better would be to compare Tel4.01 and Tel(da) – both ds with 3´overhangs, Tel(da) with 7-deaza-8-aza-dG not able to form G4.

·         I wonder if it is possible to compare results presented in Figure 1 – loss of Tel4.01 binding to Flag-hnRNP U FL in the LiCl buffer inhibiting G4 formation – and Figure 3 – binding of Flag-hnRNP U FL to Tel(da) containing 7-deaza-8-aza-dG and thus not forming G4 structure?

·         Did you verify that the PCR quantification of Tel linker oligo bound to Tel4.01 3´overhang or ssTel (Figure 4) reflects the amount of respective oligonucleotide (by, e.g., construction of calibration curves using samples with different concentrations of Tel4.01 or ssTel mixed with Tel linker as templates)?

·         The same verification of qPCR is recommended for the assay quantifying the accessibility of T24G21 and T24RG21 oligos bound to Flag-hnRNP U 683C for ExoI digestion (Figure 5). The conclusion that the partial protection of T24G21 oligo against degradation by Exo1 is due to stabilization of G4 structure by Flag-hnRNP U 683C binding should be supported by performing this assay with fragment of hnRNP U which is not able to bind to telomeric repeats (based on data presented in Figure 2).

·         The commentary to the Figure 6 was not clear. The goal of this experiment was to present, that binding of Flag-hnRNP U 683C to ssTel prevented RPA2 binding, but in the text (l. 273-274) the impossibility of ssTel(da) to bind Flag-hnRNP U 683C was commented.

·         I would recommend to present that hnRNP U promotes G-quadruples formation / stability, not activity (l. 284, 304).

·         The sentence “It also promotes telomeres from nuclease degradation.” (l. 286) is of biological relevance or based on Figure 5 data only?

·         How the fact that hnRNP U is directly associated with telomeres implies its direct roles in transcription, RNA processing and protein degradation (l. 299-301)?

·         Could you specify your hypothesis that “…hnRNP U explains why telomeres are S/MARs” (l. 309-310)?

Author Response

Reviewer 1

1.     There was stated in the Abstract (l. 17-18) that “new methods confirmed by the standard method” were used. Which of methods listed (l.18-19) are considered as new and which as standard ones?

We noticed that it is difficult for the reader to understand, so we omitted the phrase.

2.     Conclusions in Abstract, l. 21-24, should be re-phrased to be clear per se (without reading results). 

We have rewritten so that readers can understand the content of the paper by its abstract.

3.     When the sentence in the Intro is introduced by “We have suggested that…” it is expected that the work of authors of the ms will be presented. This is not the case here, l. 38-40.

We omit the ”we have suggested”

4.     G-quadruplexes are presented as structural motives regulating access of telomeres for telomerase (l. 44-46). In this context, other secondary telomere DNA structures, as t-loops and D-loops, should be mentioned. 

Now we add an explanation of the secondary telomere DNA structure.

5.     How cDNA fragments were constructed by restriction enzymes digestion (l. 77-79), what was the template for this digestion?

Full length of cDNA was obtained from Human Universal QUICK-Clone II (Clontec) as a template. With this cDNA, Flag-N704, Flag-200-472, Flag-474-704, and Flag-683C were obtained by specific restriction enzymes (as described in the text) as a templete. With this cDNA, Flag-N704, Flag-200-472, Flag-474-704, Flag-683C were obtained by specific restriction enzymes, as now described.

6.     RT-PCR (l. 122) is a standard abbreviation for reverse transcription followed by PCR. For quantitative PCR, I would recommend qPCR. 

We have changed to qPCR.

7.      I was rather confused by the structure of oligonucleotides used for binding assays (Chapter 2.9.). Am I right that Tel4.01 is ds oligo with 3´overhang, Tel(ds) ds oligo without 3´overhang and Tel(da) ds oligo containing 7-deaza-8-aza-dG with 3´overhang? I would recommend to add this information when introducing respective experiment in the Results section.

Please see the now included Figure 3A.

8.     There is stated in the text that LiCl-containing buffer (G4-inhibiting conditions) did not inhibit the binding between anti-Flag and Flag-hnRNP U (199-200) but there is no comment on the loss of binding of Tel4.01 and Flag-hnRNP U FL under these conditions, which is, in my opinion, the crucial result of this experiment.

Interaction between Tel4.01 and Flag-hnRNP U FL was decreased under LiCl-containing buffer but not under KCl-containing bufferSpecifically, LiCL-buffer Tel4.01 and Flag-hnRNP U FL binding diminishes (Fig. 1) but also LiCL-buffer affects Flag-hnRNP U FL, and diminishing antibody gives same result.

9.      The statement “Oligo alone” in Figure 1 tables is rather confusing because interpretation that oligo was used in the assay instead of respective protein is possible.

We have now explained –

Figure 1. We did not add protein, only beads and oligo (negative control)

Figure 5. Oligo only (input)

In Figure 1B (spot 5), 1C (spot 3) and 2B (spot 5), 5% of Tel 4.01 used in the binding assay was spotted as an input.

10.   Authors compared binding of Tel(da) and Tel(ds) oligos to Flag-hnRNP U FL (l. 230-231, Figure 3). If I understood well the structural specificities of respective oligos, better would be to compare Tel4.01 and Tel(da) – both ds with 3´overhangs, Tel(da) with 7-deaza-8-aza-dG not able to form G4.->

We understood that to increse specificities of respective oligo, it would be better to compare Tel4.01 and Tel(da), both ds with 3 overhangs, Tel(da) with 7-deaza-8-aza-dG is not able to form G4.We are more interested to see Tel(da) and compare with Tel4.0/Tel than compare with Tel(ds)/dsTel. We should do the experiment but we have no time. In addition, we should compare Tel(da) with Tel4.01/Tel rather than Tel(ds)/dsTel, and include Tel och Tel(da), but, again, we have no time

11.   I wonder if it is possible to compare results presented in Figure 1 – loss of Tel4.01 binding to Flag-hnRNP U FL in the LiCl buffer inhibiting G4 formation – and Figure 3 – binding of Flag-hnRNP U FL to Tel(da) containing 7-deaza-8-aza-dG and thus not forming G4 structure?

We can quantitatively compare how much hnRNP U which binds to respective oligo, or how much of respective oligo which binds hnRNP U. It would be an interesting experiment, but we have no time.

12.    Did you verify that the PCR quantification of Tel linker oligo bound to Tel4.01 3´overhang or ssTel (Figure 4) reflects the amount of respective oligonucleotide (by, e.g., construction of calibration curves using samples with different concentrations of Tel4.01 or ssTel mixed with Tel linker as templates)? -> 

Every result was evaluated by calibration curve, and bound oligo was ascertained with qPCR and plotted, then converted to Mol (0 is 1).

13.    The same verification of qPCR is recommended for the assay quantifying the accessibility of T24G21 and T24RG21 oligos bound to Flag-hnRNP U 683C for ExoI digestion (Figure 5). The conclusion that the partial protection of T24G21 oligo against degradation by Exo1 is due to stabilization of G4 structure by Flag-hnRNP U 683C binding should be supported by performing this assay with fragment of hnRNP U which is not able to bind to telomeric repeats (based on data presented in Figure 2). 

Every result was evaluated by calibration curve, and bound oligo was ascertained with qPCR and plotted, then converted to Mol (0 is 1), then we set ExoI(-)=1.

14.    The commentary to the Figure 6 was not clear. The goal of this experiment was to present, thatbinding of Flag-hnRNP U 683C to ssTel prevented RPA2 binding, but in the text (l. 273-274) the impossibility of ssTel(da) to bind Flag-hnRNP U 683C was commented. 

Flag-hnRNP U ssTel bound to hnRNP U, but ssTel(da) did not. The result suggests that hnRNP U competes with binding of RPA2 and promotes binding of ssTel.

15.    I would recommend to present that hnRNP U promotes G-quadruples formation / stability, not activity (l. 284, 304). 

Thank you, we changed to ’stability’.

16.    The sentence “It also promotes telomeres from nuclease degradation.” (l. 286) is of biological relevance or based on Figure 5 data only?

We think the result of exonuclease protection assay shows a relevant result since we have repeated the experiment and got similar results.We think that Figure 5 is enough.

17.    How the fact that hnRNP U is directly associated with telomeres implies its direct roles in transcription, RNA processing and protein degradation (l. 299-301)?

We have read papers showing that telomeres have many functions.Nucleic Acids Res.2009 Jul;37(12):3840-9. doi: 10.1093/nar/gkp259. Sci Rep.2019 Feb 8;9(1):1701. doi: 10.1038/s41598-018-38297-6.

However, these references were not important for this article and were therefore omitted.

18.    Could you specify your hypothesis that “hnRNP U explains why telomeres are S/MARs” (l. 

T 309-310)?

We think telomeres are S/MARs, provided they are located at the distal ends of chromosomes and thus are equipped with single-stranded 3’ overhangs. Being a structural constituent of the nuclear matrix, specifically recognizing telomeric ssDNA, hnRNP U will provide a possible explanation as to why telomeres are S/MARs.

Reviewer 2 Report

Telomeres and G- Quadruplex are regulated by binding protein hnRNP

The study of Izumi et al describes the role of hnRNP U in telomere biology.

Although well designed, there are several issues that are needed to be addressed before publications.

The title should be improved: Telomere function and the G-quadruplex formation are regulated by…. Maybe?

Line 17 abstract: we used new methods confirmed by the standard methods for what???

Line 22: dependent on the amount of E coli DNA?

Line 33: TTAGGG sequences are not so conserved among species- only in vertebrates this is the sequence!

Line 35: the authors have to explain – being recognized as broken DNA ends that are needed to be repaired by DNA repair mechanisms.

Somewhere in the introduction section it should be noted that telomerase is specific mainly for cancer cells

Line 51- spell out- RPA and POT1.

Line 57- …are needed in vivo for what? Please explain!

Line 61: please explain transcriptional regulation- based on what?

Line 70- spell out MEFs.

Before the results: the authors should add a cartoon of the FLAG- full length and the FLAG-N-704.

There are many many mistakes in English throughout the manuscript. Examples include: "subjected by Western Blotting?" "by competition with E coli?" and many more.

You cannot write The same treatment as in Figure 1A, last sentence.

line 223 – should be better explained.

Figures 4, 5 – no ordinate titles.

It is not clear based on what the authors claim that T24G21 is known to form G- quadruplexes.

Rephrase headline 3.3

Rephrase lines 270-272, 284

Line 302- conform?

Line 306- what is S/MARs? Please spell out.

In the discussion, the authors should refer to the fact that their study was conducted with oligo nucleotides which can only partially reflect the in vivo situation.

Author Response

Reviewer 2

Telomeres and G- Quadruplex are regulated by binding protein hnRNP

The study of Izumi et al describes the role of hnRNP U in telomere biology.

Although well designed, there are several issues that are needed to be addressed before publications.

The title should be improved: Telomere function and the G-quadruplex formation are regulated byhnRNP U…. Maybe?
We have changed the title to (Telomere function and the G-quadriplex formation are regulated by hnRNP U). Thanks for the suggestion!

Line 17 abstract: we used new methods confirmed by the standard methods for what???
We understand your concern, it was a mistake and we omitted the phrase.

Line 22: dependent on the amount of E coli DNA?
The text was omitted.

Line 33: TTAGGG sequences are not so conserved among species- only in vertebrates this is the sequence!
We changed the phrase to the below:
Telomeric DNA is highlyconserved between species (vertebrates).

Line 35: the authors have to explain – being recognized as broken DNA ends that are needed to be repaired by DNA repair mechanisms.
The essential role of telomeres is to protect chromosome ends from recombination and fusion. We omitted ”and from being recognized as broken DNA ends, that are needed to repaired by DNA repair

Somewhere in the introduction section it should be noted that telomerase is specific mainly for cancer cells.

Telomerase expression is a common feature of cancers, although some cancers use non-telomerase dependent alternative lengthening of telomeres (ALT). However, we focus only on in vitroexperiments.

Line 51- spell out- RPA and POT1.
We havedone that on Line 56.

Line 57- are needed in vivo for what? Please explain!
We suppose that about half of the telomeric overhangs are bound by RPA. Weomitted the phrase which is ambiguous.

Line 61: please explain transcriptional regulation- based on what?
The nuclear matrix-associated hnRNP U/SAF-A protein has been implicated in diverse pathways from transcriptional regulation to telomere length control and to X inactivation. However, the precise mechanism underlying each of these processes has remained elusive. 
We have included the reference 
Mol Cell.2012 Mar 9;45(5):656-68. doi: 10.1016/j.

Line 70- spell out MEFs.
(
Line 81)Mouse Embryonic Fibroblast

Before the results: the authors should add a cartoon of the FLAG- full length and the FLAG-N-704.We have drawn thiscartoon at top of Figure 1A.

There are many many mistakes in English throughout the manuscript. Examples include: "subjected by Western Blotting?" "by competition with E coli?" and many more. 
Thank you! Wehave correctedas much as we could.The Englishtexthas now been checkedby an external linguist.

You cannot write The same treatment as in Figure 1A.
 last sentence. Text added to several Figures, ”Same treatment as in Figure 1A”

line 223 – should be better explained.(line 215 now) 

The binding assay showed that hnRNPU FL bound Tel 4.01, but hnRNPU N704 did not, suggesting that the C-terminus of hnRNP U binds Tel 4.01 and G-quadruplex (Figure 1B). The same binding assay was done with Flag-hnRNP U FL with Tel 4.01 under G-quadruplex promoting (100 mM KCl) and inhibiting (100 mM LiCl) conditions.  

Figures 4, 5 – no ordinate titles. 
Y axis titles are now given both in graphs and in Figure texts.

It is not clear based on what the authors claim that T24G21 is known to formG-quadruplexes.

In this case, the G-quadruplexformsoligonucleotide (T24G21)but a control origonucleotide is unable to form (T24RG21).  Please read https://www.ncbi.nlm.nih.gov/pmc/articles/PMC1888815/  Yuan Yao et al.

Rephrase headline 3.3 
We changed to
3.3. Binding assay of Flag-hnRNP U683C and Tel or ssTel. Competition with TERRA

Rephrase lines 270-272, 284  (Line 275-278)
IPs were mixed with Tel or ssTel oligonucleotides, followed by adding indicated-fold (Figure above) amount of TERRA oligonucleotides. After washing the beads, adapter oligonucleotide (Tel linker) was mixed and bound adapter oligonucleotide was detected by qPCR.

This result indicates that the hnRNP U 683 C, but not the control (Exol-), would protect the G-quadruplex formation from Exonuclease I.

Line 302- conform?
changed to ”show

Line 306- what is S/MARs? Please spell out.
Scaffold/matrix attachment regions

In the discussion, the authors should refer to the fact that their study was conducted with oligo nucleotides which can only partially reflect the in vivo situation. 

Yes, we dothis!  Thanks somuch!

Reviewer 3 Report

This manuscript provides some biochemical insight into the function of hnRNP U in the context of telomere biology. This reviewers has the following the following recommendations:

1) Introduction is not of sufficient quality. The question should be clearly stated. Why is hnRNP U being studied?

2) A figure illustrating the telomerase complex would be helpful.

2) For the hnRNP U construct, is the sequence from human? Not specified in the text.

3) Please specify the difference between ssTel and ssTel(da)

4) Line 283. When was the PC3 cell line used?

5) Line 143. Title should describe the experiment, not the result.

6) Font size is not consistent throughout the text. Grammatical errors and typos are observed.

7) Line 316. This is not an adequate conclusion. Conclusion is vague and not supported by the data.

Author Response

Reviewer 3

This manuscript provides some biochemical insight into the function of hnRNP U in the context of telomere biology. This reviewers has the following recommendations:

1) Introduction is not of sufficient quality. The question should be clearly stated. Why is hnRNP U being studied?
We have changedthe text to get more focus to hnRNPU.

2) A figure illustrating the telomerase complex would be helpful.
We have now inserted such as figure.

2) For the hnRNPU construct, is the sequence from human? Not specified in the text.
Yes, we specifiedHuman Universal QUICK-CLONE II

3) Please specify the difference between ssTel and ssTel(da)
Wnow show an illustration.

4) Line 283. When was the PC3 cell line used?
It was mistake to write PC3,we did not use the cell line.Thank you for pointing this out!

5) Line 143. Title should describe the experiment, not the result.
We changed Title.

6) Font size is not consistent throughout the text. Grammatical errors and typos are observed.         We havechecked the paper and tried to correct. The text has now been read by an English linguist.

7) Line 316. This is not an adequate conlusion. Conclusion is vague and not supported by the data.
We have re-writtenthe conclusion.Thank you!

Round 2

Reviewer 1 Report

In the revised version, Izumi and Funa made relevant improvements of the ms. Nevertheless, in this referee’s opinion, suggested experiments which were in authors´ response commented by „we have no time“ would significantly increase reliability of presented data. Similarly, it will be appreciated if standard curves (related to data presented in Figures 4, 5) which have been constructed (as stated in authors´ comments) are presented either as supplementary data or as an additional information for the referee.

Minor comments:

·         I recommend not to use ambiguous abbreviations in Abstract (Tel, ssTel), please, introduce them properly.

·         Although information about other secondary structures formed by telomeres (except of G4) were added based on my recommendation (l. 55-57), the formulation is imprecise  -  please, delete it (rather than re-phrase).

·         Lines 52-53 – “The G-quadruplex is formed in nucleic acids, which are rich in guanines, and it is known to have a secondary structure. “ – this does not make sense .

·         Line 66 – POT1 and TPP1 are presented as ss telomere DNA binding proteins. This is valid for POT1 only, TPP1 forms a bridge between ss and ds part of telomere via binding POT1 and TIN2 (TIN2 binds to TRF1 and TRF2).

·         Line 91 – “as described by us”  - add reference, please.

·         Line 225 and 238 (Figure 1 legend): “These results are consistent to Figure 1A” and “Same treatment as in Figure 1A”, respectively – but Figure 1A is a schematic illustration of FL and N704 Flag-hnRNP U protein.

Author Response

Thank you very much for your useful review. We have followed your proposal to show standard figures as a Supplement — please see Supplemental Figures A and B.